# Enhancing rice production sustainability and resilience via reactivating small water bodies for irrigation and drainage

Sisi Li [1,2,3], Yanhua Zhuang[1,2,3], Hongbin Liu[4], Zhen Wang [5,6], Fulin Zhang[7], Mingquan Lv[8], Limei Zhai[4], Xianpeng Fan[7], Shiwei Niu[9], Jingrui Chen[10], Changxu Xu[10], Na Wang[9], Shuhe Ruan[1,2,3], Wangzheng Shen [1,2,3], Menghan Mi[1,2], Shengjun Wu[8], Yun Du[1,2,3] & Liang Zhang [1,2,3] ✉

Rice farming threatens freshwater resources, while also being increasingly vulnerable to drought due to climate change. Rice farming needs to become more sustainable and resilient to climate change by improving irrigation drainage systems. Small water bodies, used to store drainage water and supply irrigation in traditional rice farming systems have gradually been abandoned in recent decades. This has resulted in a higher water footprint (WF) associated with rice farming due to increased freshwater usage and wastewater release, also leaving rice production more vulnerable to extreme weather events. Here, we propose how protecting and reactivating small water bodies for rice irrigation and drainage can decrease rice production WF in China by 30%, save 9% of China's freshwater consumption, increase irrigation self-sufficiency from 3% to 31%, and alleviate yield loss in dry years by 2–3%. These findings show that redesigning rice irrigation drainage systems can help meet water scarcity challenges posed by climate change.

Agriculture is a major threat to water security due to its high freshwater usage for irrigation and wastewater discharge during drainage[1,2]. Rice production is particularly concerning since it consumes approximately 40% of global freshwater resources[3] and releases more wastewater than upland fields due to its low water use efficiency[4]. As the extent and intensity of irrigation and drainage systems increase, water stress from agriculture becomes more severe[5,6]. Irrigation areas have to be expanded by more than 5% in warm regions to offset warming-induced production losses by the 2050s, which demand more water resources[7]. Extreme weather increases agricultural nutrient losses and the harmful effects on water quality[8]. This increasing water stress, in terms of both quantity and quality, threatens the sustainability of food production and global food security[9]. To achieve the United Nations' sustainable development goals of ending hunger (Target 2.1), promoting sustainable agriculture (Target 2.4), improving water quality (Target 6.3), and increasing water use efficiency (Target 6.4) by 2030, both on-farm level technologies and agricultural system transformation are needed[10].

[1]Hubei Provincial Engineering Research Center of Non-Point Source Pollution Control, Innovation Academy for Precision Measurement Science and Technology, Chinese Academy of Sciences, Wuhan 430077, PR China. [2]Key Laboratory for Environment and Disaster Monitoring and Evaluation of Hubei, Wuhan 430077, PR China. [3]University of Chinese Academy of Sciences, Beijing 100049, PR China. [4]Institute of Agricultural Resources and Regional Planning, Chinese Academy of Agricultural Sciences, Beijing 100081, PR China. [5]State Environmental Protection Key Laboratory of Soil Health and Green Remediation, Huazhong Agricultural University, Wuhan 430070, PR China. [6]Interdisciplinary Research Center for Territorial Spatial Governance and Green Development, Huazhong Agricultural University, Wuhan 430070, PR China. [7]Institute of Plant Protection, Soil and Fertilizer Sciences, Hubei Academy of Agricultural Sciences, Wuhan 430064, PR China. [8]Chongqing Institute of Green and Intelligent Technology, Chinese Academy of Sciences, Chongqing 400714, PR China. [9]Liaoning Academy of Agricultural Sciences, Shenyang 110161, PR China. [10]Institute of Soil & Fertilizer and Resources & Environment, Jiangxi Academy of Agricultural Sciences, Nanchang 330200, PR China. ✉e-mail: lzhang@apm.ac.cn

While on-farm technologies such as water-saving irrigation, controlled drainage, and prioritized fertilization are relatively well-studied[11–15], there is less research on irrigation drainage system transformation[16]. A recent study in the Ganges Basin showed how irrigation system transformation can be cost-effective in regulating water resources between monsoon and dry seasons[17]. Castellano et al. discussed how drainage system redesign can mitigate nutrient losses, increase fertilizer use efficiency and reduce greenhouse gas emissions[5]. For resilience, modern water storage strategies call for an integrated system including reservoirs, ponds, tanks, aquifers, and wetlands[18]. These studies tackle one or two aspects of irrigation drainage system transformation, i.e., enhancing water resource regulation, reducing the water quality impact of drainage, or improving system resilience to extreme weather. A more comprehensive analysis and solution addressing the increasing water stress, both in terms of sustainability and resilience, is still quite limited.

Looking back at the development history of irrigation drainage systems may provide insight into future solutions. Ponds (or tanks) played an important role in ancient times, especially in rice-growing regions such as China and India[19–21]. A typical pond-based system in China named Beitang (Fig. 1a) with more than 2500 years of history is regarded as a paradigm for balancing rice production and environmental impact[21]. This system is considered to be decentralized because

small ponds are scattered throughout the system, serving as temporary rainfall and runoff storage tanks as well as main irrigation sources. By contrast, modern irrigation drainage systems are centralized relying on large reservoirs, pumping stations, and canals, in which scattered small ponds are largely lost or bypassed so that the management practices become more unified relying on remote freshwater supply (Fig. 1b). Unlike the modern system that 100% rely on remote freshwater resources for irrigation, the traditional decentralized system is partly self-sufficient since ditches and ponds within the system provide local irrigation supply. In addition, small water bodies (ditches and ponds) are wetland ecosystems with disproportionally high value in nutrient retention and biodiversity support[22–24]. Given the increasing water and food stress worldwide, referring to ancient wisdom in utilizing small water bodies may help develop improved water management approaches for addressing current and future climate change challenges[25].

The present study took China as a case to explore pathways toward more sustainable and resilient rice irrigation and drainage. First, a systematic survey was conducted to examine the evolution and current status of rice irrigation drainage systems, especially how small water bodies are used. Then, water quantity and quality modeling by WQQM-PIDU model[26] using aforementioned survey data along with various climate data and nutrient processing parameters[27] were

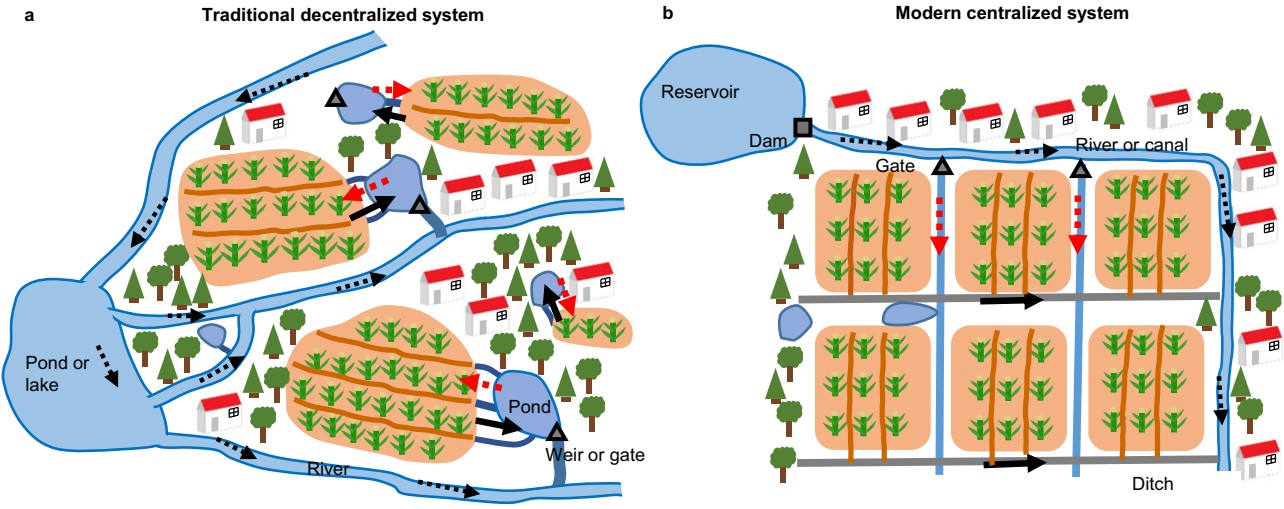

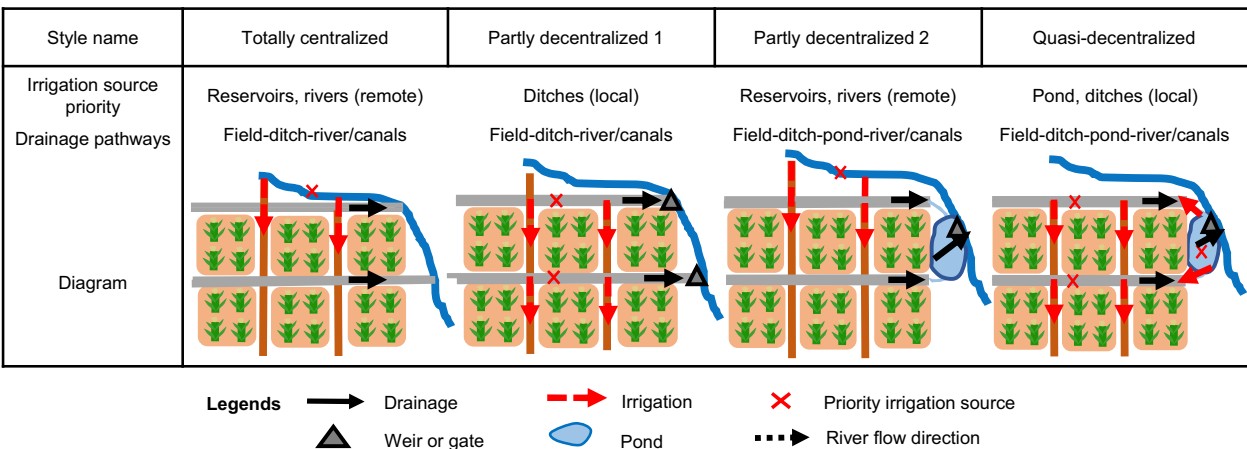

**Fig. 1 | Diagrams of different irrigation drainage systems. a** Traditional decentralized system and **b** modern centralized system on the district scale. **c** Four irrigation drainage unit (IDU)-scale management styles within the modern system. The Traditional decentralized system in panel **a** is drawn based on the Beitang system in China referred to ref. 21, small ponds are scattered in the system and almost all IDUs use small ponds to store drainage water and supply irrigation.

performed to determine and compare the water impact as well as system resilience of differently managed systems. The water impact was quantified by the water footprint (WF) of rice production[28] while the system resilience was quantified by irrigation self-sufficiency and the potential alleviation of yield loss due to water stress in dry years[29]. Based on the benefits of the decentralized-managed systems, practical approaches were proposed to reactivate small water bodies for irrigation and drainage within the context of modern systems. The potential benefits, costs, and implementation feasibility of these approaches were analyzed and discussed to provide promising pathways for rice irrigation drainage system transformation in China to address current and future climate challenges.

## Results

### Abandoned ponds in China's irrigation drainage systems

The irrigation drainage systems in China have changed fundamentally in recent decades (Fig. 2a). In the 1950s, 8.3 million decentralized Beitang systems served 39% of the total irrigated area. This number shrunk to 2.2 million in 2006 and slightly recovered to 3.4 million in 2016. Although the traditional system was losing its number and area, the area of irrigated fields and drainage fields in China increased continuously due to the rapid construction of reservoirs, canals, and pumping stations. This evolution enhances the efficiency of water resource regulation. Concurrently, it has led to a significant loss, disconnection, or disuse of scattered ponds, which were mainly filled for farmland[30] to feed the fast-growing population before 1980, and used for intensive aquaculture production afterward. Ponds were also disused due to poor maintenance[21] resulting from policy changes and a lack of financial investment (Fig. 2a and Supplementary Table 1). Currently, small ponds are mainly found in southern China, with an average area percentage of 1.2% in the Southeast coastal rice region and 1.5% for the Changjiang River basin rice region. Small water bodies, including ditches and small ponds together, make up an average of 6.1% of areas of rice production systems in China, ranging from 2.2 to 11.0% in different provinces (Fig. 2b).

Most rice regions in China currently have modern irrigation drainage systems (Fig. 1b). However, the modern system is composed of different sub-systems known as irrigation drainage units (IDUs), which are made up of rice fields, ditches and perhaps ponds, weirs or small pumping stations that are independently managed by one entity (farmer, rural cooperative organization, or enterprise). Their spatial scales range from several hectares to several square kilometers. There are four styles of IDUs based on how ditches and ponds are used (Fig. 1c): (1) totally centralized, with no ponds in drainage pathways and remote freshwater used for irrigation; (2) partly decentralized 1, with no ponds in drainage pathways and local ditch water given priority for irrigation; (3) partly decentralized 2, with ponds in drainage pathways and remote freshwater used for irrigation; and (4) quasi-decentralized, with ponds in drainage pathways that are also used as the priority source for irrigation. The quasi-decentralized sub-system is quite similar to the traditional decentralized system but remains in less than 5% IDUs currently, while the totally centralized style is dominant, comprising over 60% of current IDUs. In 80% of IDUs, ponds are not in use for rice irrigation and drainage (totally centralized and partly decentralized 2 styles), mainly due to hydrological isolation or poor maintenance. The disuse of ponds, along with the evolution of rice irrigation drainage systems in China, has influenced water resource utilization, the water environment, and the system's resilience to climate change.

### Increased water footprints with the system evolution

As ponds have been disused in current irrigation drainage systems, the water footprint (WF) of rice production has increased, because the centralized-managed system bypassing ditches and ponds has more WFs than the decentralized-managed system utilizing them (Fig. 3). On average, the WF of the quasi-decentralized system is 910 m³/ton dry

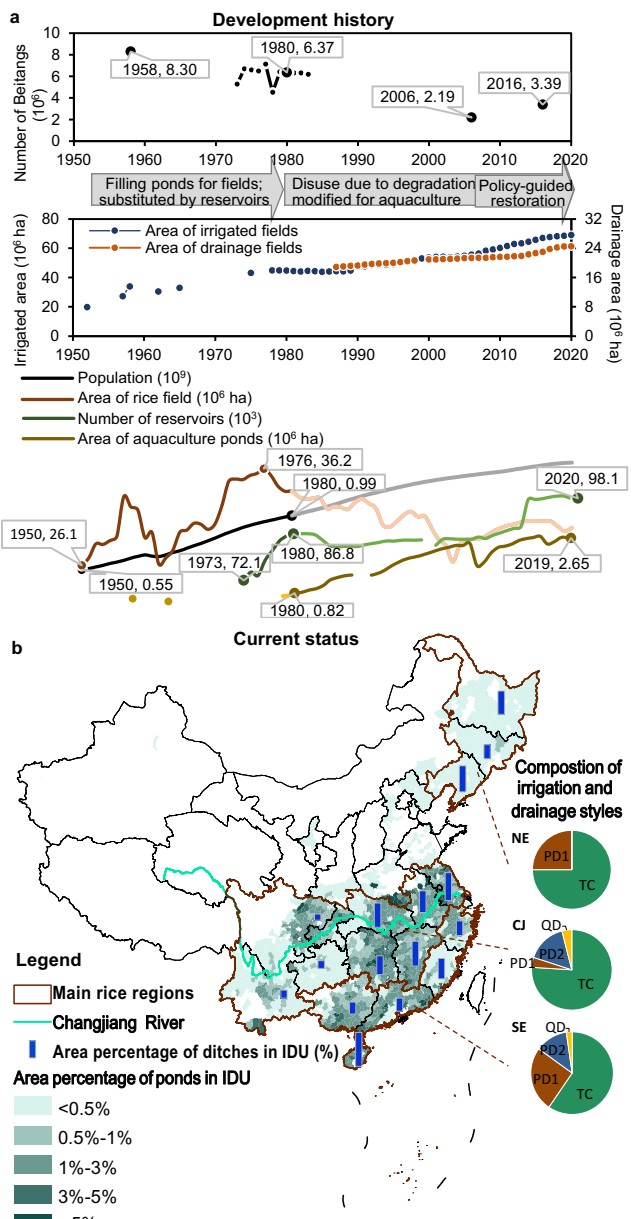

**Fig. 2 | Irrigation drainage system evolution and the current status of how small water bodies are utilized in China. a** Development history of irrigation drainage systems in China, showing the degradation of the pond-based decentralized system (Beitang) but increased areas of irrigated and drainage fields, along with related factors. Highlighted parts of lines showed the most influencing period of these factors; **b)** the current status of remaining ditches and ponds in rice-growing regions of China and their utilization styles. IDU: irrigation drainage unit. TC totally centralized style, PD1 partly decentralized 1 style, PD2 partly decentralized 2 style, QD quasi-decentralized style, the difference between the four styles is depicted in Fig. 1c. NE northeast rice region, CJ Changjiang River basin rice region, SE southeast coastal rice region. Source data are provided as a Source Data file.

matter, with a 90% probability range of 619–1491 m³/ton, which is 33% less than that of the totally centralized system (1360 m³/ton with a 90% probability range of 854–2193 m³/ton). The two systems have similar green WFs (rainwater consumption). However, the quasi-decentralized system reduces its blue WF (irrigation freshwater consumption) by an average of 23% through the recycled usage of gray water (drainage wastewater) retained in ditches and ponds instead of using blue freshwater outside of the system. Moreover, the gray WF from the quasi-decentralized system that enters surrounding surface waters is

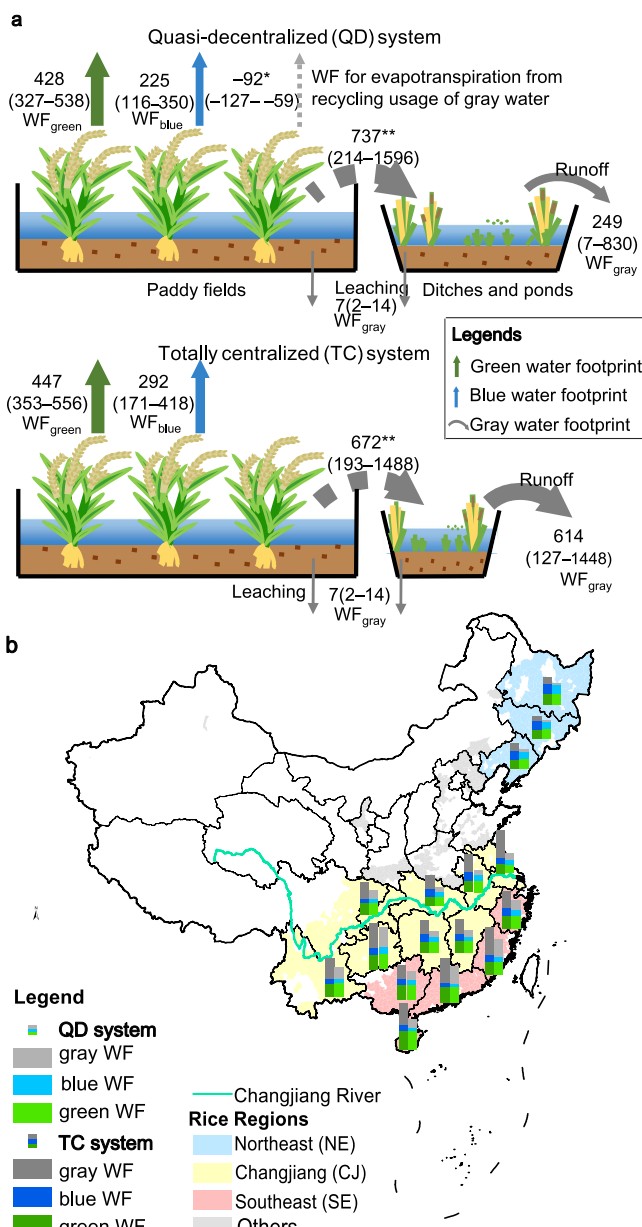

**Fig. 3 | Comparison of water footprint (WF) impact of quasi-decentralized system and totally centralized system. a** Diagram of the average WF of rice production in the whole of China for the two systems, **b** WF impact of the two systems in different rice-growing provinces. WF values in panel **a** are mean values with 90% probability ranges in brackets of the 9000 simulation results representing the variability of climate conditions and the uncertainty of nutrient retention parameters. Solid green, blue, and gray arrows are green, blue, and gray WF out of the system, in m³/ton; the dashed gray arrows indicate within system WF: * denotes WF for evapotranspiration from recycling usage of gray water, ** denotes gray WF from paddy fields to ditches and ponds. Note, the number of plants in the graph does not represent rice yield. Source data are provided as a Source Data file.

Changjiang River basin region (33–55%) and in Hainan province of the Southeast coastal region (36%) (Fig. 3b). The area percentage of ditches and ponds and their use in irrigation and drainage are key factors that determine the impacts. In fact, these rice-growing provinces (Hubei, Hunan, Anhui, Jiangxi, Jiangsu, and Hainan, province names in Supplementary Fig. 1) with greater WF differences have an area percentage of ditches and ponds greater than 7% (Fig. 2b). In addition, the gray WF contributed the most to the WF disparities between quasi-decentralized and totally centralized systems. For example, the southern provinces have a larger gray WF than the northeast provinces, so the WF disparities between different systems are also larger.

## Reduced resilience to extreme weather with the system evolution

The disuse of ditches and ponds for irrigation and drainage has also reduced the system's resilience to extreme weather because these small water bodies provide flexible irrigation independently. In fact, irrigation self-sufficiency (defined as the percentage of irrigation water from local ditches and ponds) is largely determined by the local storage volume provided by ditches and ponds (Fig. 4). Decentralized-managed systems with a local storage volume of less than 35 mm field water have irrigation self-sufficiency ranging from 4 to 39%. When the local storage volume increases to 35–55 mm and more than 55 mm field water, irrigation self-sufficiency increases to 14–55% and 13–80%, respectively (Fig. 4a). More ditches and ponds included in drainage pathways provide more retained drainage water and rainwater to support irrigation by the system itself. Besides local storage volume, irrigation self-sufficiency also varies with time or climate conditions. As the climate becomes drier, irrigation self-sufficiency generally decreases. But decentralized-managed systems with more than 35 mm local storage volume have irrigation self-sufficiency of about 20–40% in 25% dry years (aridity index of the growing season greater than 1) or dry irrigation events (more than 7 continuous no-rain days). Even in 5% extreme dry conditions (aridity index greater than 1.6 or continuous no-rain days greater than 14 days), these systems can still supply 20–30% irrigation water independently.

In normal climate years, the local storage volume and irrigation self-sufficiency provided by ditches and ponds do not impact rice yields because they only change the irrigation sources while the quantity of irrigation remains unchanged. However, in 25% dry climates, where access to remote freshwater irrigation is restricted, the potential yield loss for centralized-managed systems without ditch and pond storage is 2.5% (range of 0.8 to 4.7%) greater than decentralized-managed systems with irrigation self-sufficiency of 15–40%. The difference in potential yield loss would further increase to 3.1% (range of 0.9–5.4%) under the 5% extreme dry climate. These results are consistent with the documented literature indicating that rice-growing areas utilizing scattered ponds for irrigation withstood droughts two weeks long, while yield reductions have been observed following several days of drought when small ponds have been lost[30]. Besides providing additional irrigation supply in dry climates, systems utilizing small ponds also provide additional resilience to water-logging disasters. The local storage volume also provides additional capacity for field drainage storage when drainage directed to rivers is limited during river flooding.

## Benefit and cost of reactivating small water bodies in irrigation and drainage systems

Given the advantages of the decentralized-managed systems utilizing small water bodies in reducing WF and improving system resilience, we proposed three system redesign approaches to integrate more ditches and ponds for rice irrigation and drainage: (1) recycling irrigation that prioritizes the use of retained ditch and pond water for irrigation (recycling irrigation) should be promoted, which turns the current totally centralized sub-system into a partly decentralized 1 sub-system

about 59% less than that from the totally centralized system. This is caused by the recycling usage of gray water for irrigation, as well as greater nutrient retention and purification in more ditches and ponds through such processes as sedimentation, plant uptake, and micro-organism consumption[31,32]. The wider variability of gray WF compared to green and blue WF is due to the uncertainty of the nutrient retention capability of ditches and ponds.

The difference in WF between quasi-decentralized and totally centralized systems was greater in provinces in the middle and lower

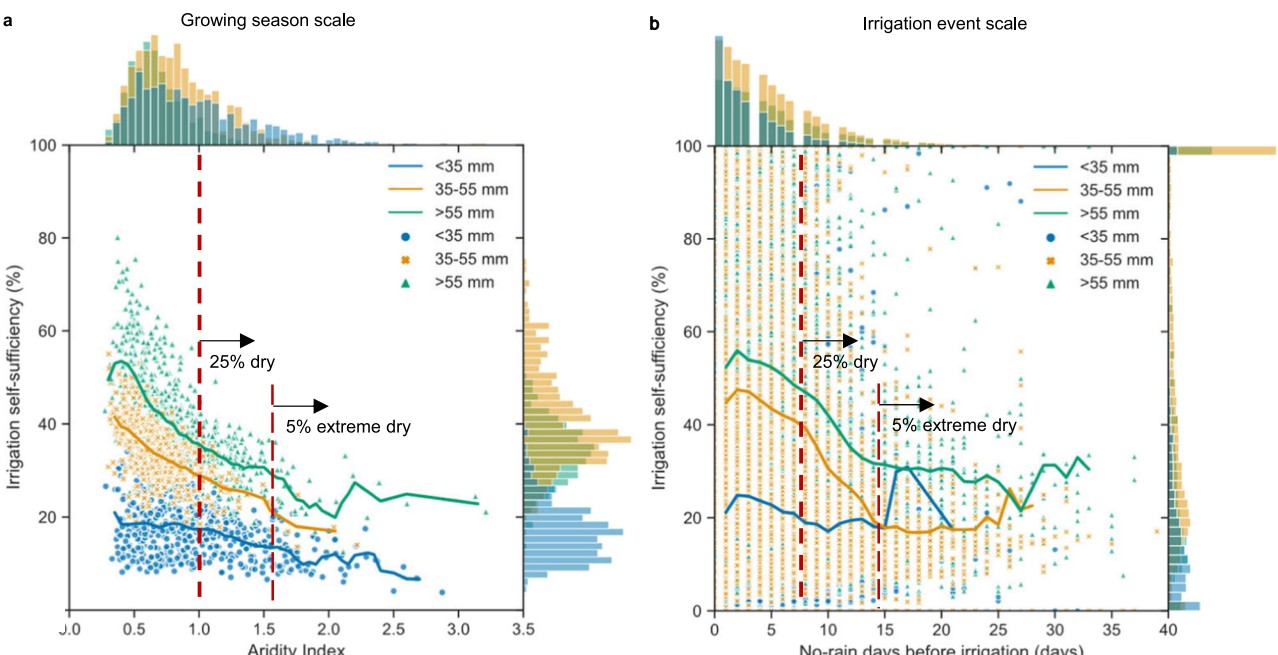

**Fig. 4 | Resilience of the quasi-decentralized system to droughts under different climate conditions. a** Irrigation self-sufficiency for the growing season scale with various aridity index; **b** Irrigation self-sufficiency for the irrigation event scale with various no-rain days before irrigation. The edge histograms show the distribution of climate conditions and irrigation self-sufficiency. The center plots show the relationship between system irrigation self-sufficiency and the climate conditions, the dots represent all different cases and the lines are the mean values for specific climate conditions. Source data are provided as a Source Data file.

and the current partly decentralized 2 sub-system into a quasi-decentralized sub-system; (2) isolated ponds and ditches should be reconnected (pond reconnection) in order to incorporate pond within drainage pathways, which turns the partly decentralized 1 sub-system into a quasi-decentralized sub-system; and (3) more ponds should be dug or dredged (pond construction) to include more ponds for decentralized irrigation and drainage. We designed a series of scenarios, from easy to difficult implementation. The easily implemented scenarios would increase the recycling irrigation percentages from current status to 100%; the moderately easy-to-implement scenarios would further conduct pond reconnection to improve the percentage of existing ponds used for irrigation and drainage to 100%; and the difficult to implement scenarios would construct ponds to increase the area percentages of ditches and ponds to 8%, the maximum allowable percentage set by the general well-facilitated farmland construction standards of China (GB/T 30600-2022).

The potential benefits of these approaches are shown in Fig. 5. The recycling irrigation approach alone would reduce the overall WF of rice production in China from 1299 to 1018 m³/ton. In addition to recycling irrigation, the pond reconnection approach would further reduce the WF to 910 m³/ton. By using the pond construction approach, the rice production WF would be further reduced to 802 m³/ton, which would be 62% of the current status. In addition to enhancing water use sustainability, the proposed approaches would also increase the irrigation self-sufficiency of rice systems in China from the current 3 to 21% for the recycling irrigation approach, 31% for the recycling irrigation plus pond reconnection approaches, and 41% for all three approaches fully implemented. This would alleviate the yield loss due to water stress under the 5% extreme dry climate from 11.3 in current status to 9.1% for the recycling irrigation approach, 8.6% for the recycling irrigation plus pond reconnection approaches, and 7.7% for all three approaches fully implemented (Fig. 5b).

The main trade-off of the proposed approaches lies in the usage of ponds that require additional land, which starts with the pond construction scenarios (Fig. 5). Without this trade-off, reactivating small water bodies currently remaining in rice irrigation drainage systems via recycling irrigation and pond reconnection approaches would reduce the WF to 910 m³/ton, which is equivalent to about 80 billion m³ of freshwater saved, or 9% of the total freshwater consumption of China[33]. Furthermore, the redesigned system would alleviate 2–3% yield loss in dry climates due to enhanced irrigation self-sufficiency. Beyond this benefit, the pond construction approach would further reduce the WF by 12% (Fig. 5a) but its benefit on yield loss alleviation under extreme dry climate is quite limited (Fig. 5b) with a cost of land occupation. If all newly constructed ponds were converted from previous rice fields, it would occupy at most 2.1% of the current rice fields or reduce 2.0% of national rice yields in normal climate years. Hence, the pond construction approach is not cost-effective in terms of system resilience in general and should be considered only when water sustainability is the priority concern.

The decision on to what extent the system redesign approaches should apply may vary among regions and provinces (Fig. 5c, d). The main challenge in the northeast rice region is resilience to extreme weather, while water sustainability is of greater concern for the southern rice regions. The recycling irrigation approach is worth implementing in most rice-growing provinces since it is the most effective in enhancing both system resilience and sustainability. Pond reconnection would be a beneficial approach in terms of water sustainability for provinces having more than 2% ponds such as Hainan and Hunan provinces. For rice-growing provinces with limited ditches and ponds currently, such as Guizhou, Yunnan, and Sichuan in the Changjiang River basin region (in which the area percentage of ditches and ponds are less than 4%), the pond construction approach would be most effective and should be considered. Its benefit on sustainability (additional WF reduction by about 50%) and resilience (additional 2% yield loss alleviation in a dry climate) might outweigh the cost in land occupation and about 5% yield reduction in normal years.

**System redesign cases and benefits confirmed by observations**
To test the applicability and benefits of the proposed system redesign approaches in practice, we chose three typical centralized-managed IDUs in the Northeast rice region, one-season and two-season rice

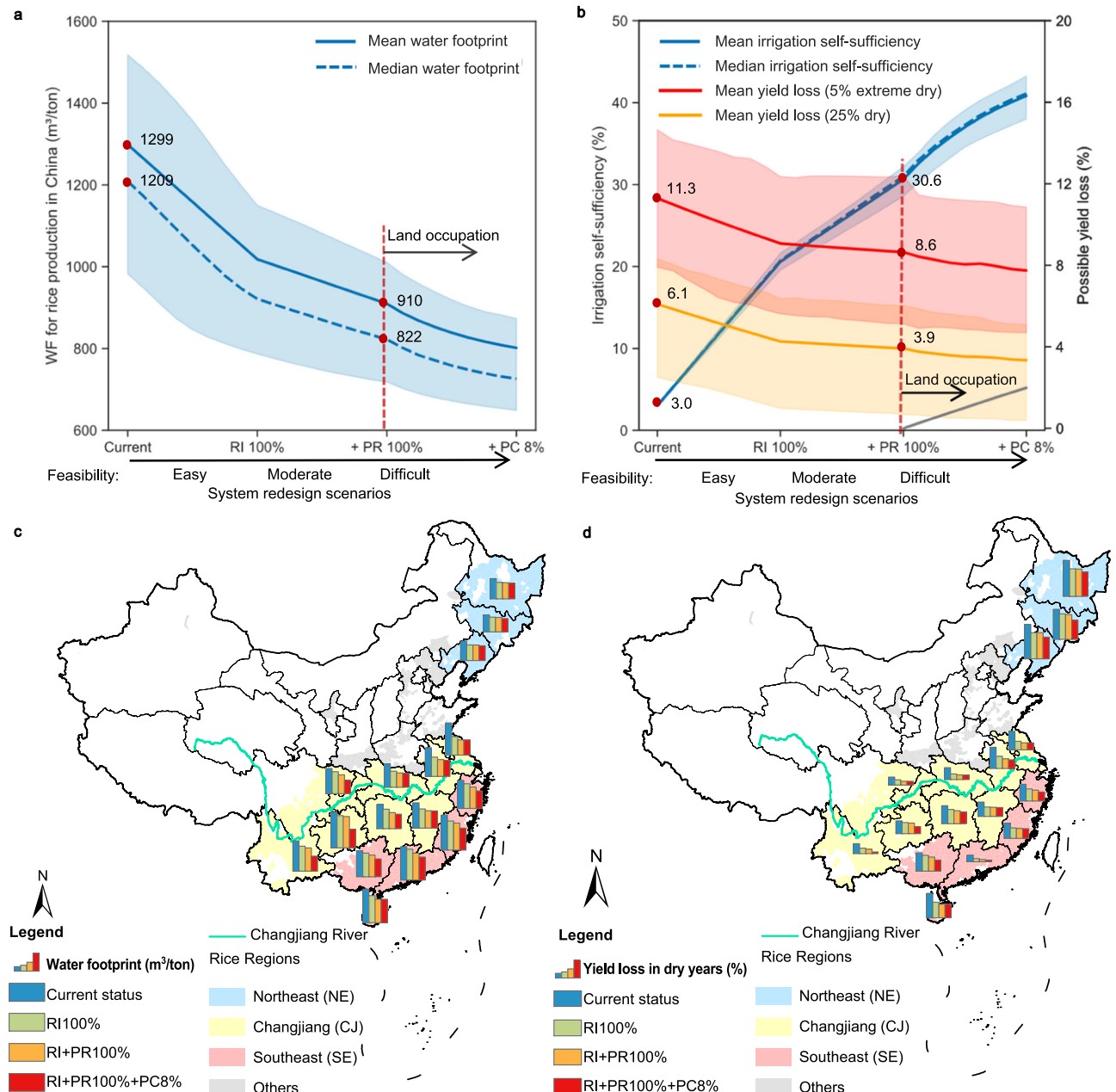

**Fig. 5 | Benefit of irrigation drainage system redesigns scenarios from easy to difficult implementation. a** Water footprints (WFs) of rice production in the whole of China, **b** Irrigation self-sufficiency and yield loss due to water stress under 25% dry and 5% extreme dry climate in the whole of China, **c** WFs for different provinces, **d** yield loss due to water stress under 25% dry climate for different provinces. The scenarios start from the current status of irrigation and drainage management styles, proceed to a system redesign approach by increasing recycling irrigation (RI) to 100% and pond reconnection (PR) to 100%, followed by the implementation of pond construction (PC) until the area percentage of ditches and ponds reaches 8%. The solid line and dashed line in panels **a**, **b** are the mean and median values respectively of 9000 simulation results representing various climate conditions and nutrient retention capabilities, the shaded areas are the inter-quantile ranges. The red dots in panels **a**, **b** signal the benefit of water sustainability and system resilience under no apparent land occupation trade-off. Source data are provided as a Source Data file.

regions in the Changjiang River Basin as cases to incorporate more ditches and ponds in irrigation and drainage (Table 1). Field observations confirmed that the redesigned IDUs with more decentralized management dramatically reduced the gray WF (65–85%) compared to the control IDUs, while the rice yield per unit area slightly increased (1.3–3.3%). There was also a reduction of fertilizer usage by more than 10% for these redesigned IDUs, which did not harm rice yield because the drainage water recycled by ditches and ponds for irrigation is generally more nutritious than remote freshwater, and this additional nutrient supplemented the reduced nutrient input from fertilizers. The reduction of fertilizer usage decreased nutrients exported to the water

environment in addition to irrigation drainage system redesign approaches, resulting in a greater gray WF reduction than simulated in Fig. 3. In this sense, the simulated benefits on WF reduction and rice yield stabilization by system redesign approaches in Fig. 5 is conservative, and more benefits can be achieved if fertilizer usage is reduced along with irrigation drainage system redesign.

## Discussion

The past decades in China witnessed fast development and construction of "big and powerful" irrigation facilities (reservoirs, canals, pumping stations), which greatly increased the areas of irrigated fields

**Table 1 | Observations in typical redesigned irrigation drainage units (IDUs) reactivating small water bodies and nearby control IDUs bypassing them in different rice regions**

| Rice region<br>Location | CJ1S<br>Anlu, Hubei | CJ2S<br>Gaoan, Jiangxi | NE<br>Panjin, Liaoning |
|---|---|---|---|
| **Redesign approaches** | RI, PR | RI, PR, PC | RI |
| **Redesigned IDU*** | | | |
| Area (ha) | 85.4 | 22.0 | 16.0 |
| Rice yield (ton/ha) | 8.50 | 13.58 | 13.25 |
| N fertilizer usage (kg/ha) | 137 | 302 | 210 |
| P fertilizer usage (kg/ha) | 24 | 60 | 39 |
| TN loads out of IDU (kg/ha) | 5.16 | 1.72 | 0.67 |
| TP loads out of IDU (kg/ha) | 0.28 | 0.13 | 0.01 |
| Gray WF (m³/ton) | 303.6 | 63.4 | 25.3 |
| **Control IDU*** | | | |
| Rice yield (ton/ha) | 8.33 | 13.14 | 13.00 |
| N fertilizer usage (kg/ha) | 168 | 347 | 282 |
| P fertilizer usage (kg/ha) | 30 | 74 | 46 |
| TN loads out of IDU (kg/ha) | 6.54 | 8.50 | 4.43 |
| TP loads out of IDU (kg/ha) | 0.40 | 0.49 | 0.04 |
| Gray water WF (m³/ton) | 867.5 | 323.3 | 170.4 |
| **Redesign impacts** | | | |
| Rice yield | 2.0% | 3.3% | 1.9% |
| N fertilizer usage | −18.5% | −13.0% | −25.5% |
| P fertilizer usage | −20.2% | −18.9% | −14.3% |
| TN loads out of IDU | −21.1% | −79.8% | −84.9% |
| TP loads out of IDU | −30.0% | −73.5% | −64.1% |
| Gray WF | −65.0% | −80.4% | −85.2% |

*Note: The control IDUs are under totally centralized management, while the redesigned IDUs in CJ1S and CJ2S regions became quasi-decentralized system, and that in NE region became a partly decentralized 1 system since ponds do not exist.

*RI* recycling irrigation, *PR* pond reconnection, *PC* pond construction, *TN* total nitrogen, *TP* total phosphorus, *CJ1S* the Changjiang one-season region, *CJ2S* the Changjiang two-season region, *NE* the northeast region.

and drainage fields, contributing to the food safety of the fast-growing population[34]. However, this construction was accompanied by the loss or disuse of small water bodies due to the pursuit of food and profit or poor maintenance caused by low financial investigation on these small ecological landscapes. With social and economic growth, sustainable agriculture is of greater priority. The increasing water stress, as well as more frequent extreme weather, require the rice irrigation drainage system to adapt and evolve. The proposed approaches to reactivating small water bodies for rice irrigation and drainage are compatible with the current systems, and only IDU-scale facilities need to be redesigned. Specifically, recycling irrigation, the most cost-effective approach, needs the installation of weirs or gates at the end of IDUs and using retained water in ditches and ponds for irrigation as a priority. It does not change the irrigation quantity or frequency, so it does not require additional labor input, making it relatively easy to be adopted. The pond reconnection approach may have an economic barrier for ponds already in use for aquaculture production, while other ponds with collective ownership have no clear management

entity, hampering the maintenance and utilization of these ponds[35]. For these reasons, only small ponds with a surface area of less than 0.33 hectares were surveyed and used for scenario analysis in this study (Figs. 2–5). Pond construction costs the most due to additional land occupation. Some ongoing policies and projects (Supplementary Table 1) may remove these barriers and provide direct financial investment in the implementation of the proposed approaches. The well-facilitated farmland construction projects[36] encourage large holders or agricultural enterprises to consolidate the management of farmland to generate economies of scale[37]. This makes pond reconnection, utilization, and maintenance easier to apply and more cost-effective. In addition, the project of hydrologic connection and construction of water beauty rural areas since 2020 provides direct policy support and financial investment for pond reconnection and pond construction. But both projects lack a clear guide to the design of irrigation drainage systems specifically in terms of the utilization of small water bodies. The findings of this study (Fig. 5) indicate the necessity of protecting small water bodies remaining in current rice-growing regions and the significance of reactivating them in irrigation and drainage management. For provinces with less than 4% ditch and pond areas, constructing more ponds needs to be considered to address the increasing water and climate challenges.

The loss of small ponds in agricultural landscapes has not only occurred in China but also in other countries around the world[19,20,38]. Our study highlighted the importance of the decentralized utilization of small ponds for irrigation and drainage management, which is critically valuable for sustainable and resilient rice production in China. Similarly, pond irrigation is attracting attention in other rice regions of the world[39]. For example, among various potential solutions to India's groundwater resource depletion issue, the rehabilitation of traditional pond (tanks in India) irrigation systems is regarded as a feasible option for its relatively low cost[38]. Recent studies in Iran and Nepal also showed that pond irrigation as a supplement increased water availability and crop productivity and relieved drought pressure[40,41]. In normal climate conditions, reservoir-based modern systems are adequate for water resource regulation. However, in extreme drought or flood, agricultural needs may not be guaranteed when competitive water management targets are urgent for other sectors. At this time, scattered small ponds provide greater flexibility and indispensable complementary flood regulation and irrigation supply functions[32,42,43]. The increased system resilience is vitally important since human-induced climate change has increased the likelihood of extreme weather and the occurrence frequency of droughts in current wet regions[44].

In terms of water sustainability, ponds in decentralized systems serve as temporary wetlands for the treatment and recycling usage of nutritious drainage water. This is a good solution to the frequently altered irrigation and drainage during rice growth. As the early-season drainage and late-season irrigation are exacerbated by wetter springs, summer droughts, and more intense precipitation events under the changing climate[8,45], there is a great advantage to storing nutritious drainage water in the early season, with local ditches and ponds to support late-season irrigation. Hence, the decentralized utilization of small water bodies for irrigation and drainage is an ideal solution for sustainable rice production in China. Interestingly, detention ponds has been recently used to collect and store subsurface drainage water for recycling irrigation in the Midwest of the USA, where there is similar early-season drainage and late-season irrigation need[46,47]. The idea of water resource and nutrient recycling underpinned is worth spreading in other agricultural regions with similar irrigation–drainage alteration patterns.

In summary, our study took a systematic and forward-looking view of the evolution of rice irrigation drainage systems. Through a systematic survey in China, we found small water bodies that were traditionally used to store drainage water and supply irrigation have

gradually been abandoned in favor of using the land to grow more food and make more profit. This has increased freshwater consumption and wastewater discharge and made rice yield more vulnerable to extreme weather events. Given the increasing water stress from both quantity and quality under the changing climate, we suggest protecting the remaining small water bodies and reactivating them for rice irrigation and drainage by reconnecting them with rice fields and using them to recycle drainage water for irrigation. This would decrease the WF of rice production in China by 30%, saving 9% of freshwater consumption of the entire country. This would also increase irrigation self-sufficiency from 3% to 31%, thereby alleviating 2–3% yield loss in dry years. The findings provide direct recommendations for ongoing policies and projects in China and helpful ideas for rice irrigation drainage system transformation towards sustainability and resilience to address the challenges we face with water and climate change.

## Methods

### Data collection and survey

We collected data on the development history and current status of irrigation drainage systems in rice regions of China through a systematic survey. The data was collected from three aspects: (1) statistical data that includes the number of Beitang, area of irrigated fields and drainage fields, population, rice yield on dry matter basis, the number of aquaculture ponds, etc. from 1950 to 2020, which were obtained from sources such as the national bureau of statistics (https://data.stats.gov.cn), China agriculture yearbook, China water conservancy yearbook, and Bulletin of the National Agricultural Census; (2) expert knowledge on four irrigation and drainage management styles in IDU scales (shown in Fig. 1c) that currently existed in three main rice regions; (3) remote sensing survey data of the area percentages of ditches and ponds in current rice irrigation drainage systems. For the area percentage of ponds, we extracted 1.75 million small ponds in rice-growing regions from a dataset of small water bodies in China recently published by ref. 48. Only small ponds with a surface area of less than 0.33 ha were included for analysis since large ponds are generally in use for aquaculture production or have collective ownership, which makes them less likely to be used for rice irrigation and drainage. With this extracted dataset, the area percentages of small ponds are calculated in 1156 rice production counties (Fig. 2b). Given the greater difficulty of remote sensing monitoring of small ditches than ponds in large areas, there were no datasets of ditches such as ponds in China. For the area percentages of ditches, we extracted the ditches by visual interpretation of Google Earth images in 45 typical IDUs across 16 rice-growing provinces (locations shown in Supplementary Fig. 1). Given the construction of agricultural ditches has national rules so that the variability of ditch construction within a province is relatively small, so the uncertainty introduced by the ditch survey on typical IDUs is limited.

To evaluate the sustainability and resilience of rice irrigation drainage systems, other data were collected, including daily climate data, field agricultural management information, and nutrient retention-related parameters of ditches and ponds. Their sources, spatial scales, preprocessing analysis, and their usage in the model simulation are described in Supplementary Table 2 and Supplementary Fig. 2. Climate data were collected from the China meteorological data service center (https://data.cma.cn). Thirty years (1988–2017) of daily climate data were collected from three different weather stations in each province in the main rice regions of China (locations shown in Supplementary Fig. 1), which is 90 year-sites daily data for each province for model simulation. These climate data were also used for climate condition analysis to determine the normal, 25% dry, and 5% extreme dry climate years. Field agricultural management data include initial nitrogen and phosphorus concentrations on the fertilization days and the water level management scheme for different rice-growing periods. The initial nutrient concentrations on fertilization

days were set based on their relationship with fertilizer application rate, soil pH, and soil organic matter extracted from 76 literatures with 3486 data[49]; the field water level management scheme was set to be normal management by farmers according to the survey. Nutrient retention-related parameters of ditches and ponds used in model simulation include the retention velocity for nitrogen ($vf\_N$) and for phosphorus ($vf\_P$) in nutrient spiraling theory and the equilibrium nitrogen and phosphorus concentrations ENCO and EPCO. They vary greatly, and a lognormal distribution was used to describe their occurring probability[22]. According to a study based on a literature survey in China[27], $vf\_N$ and $vf\_P$ ranged from 4.2–44 cm/d and 1.5–9.0 cm/d respectively, and ranges of ENCO and EPCO were set to be 0.4–5.0 mg/L and 0.03–1.1 mg/L respectively. These literature data were fitted to a lognormal probability distribution, and a Monte Carlo sampling was conducted to generate 100 sets of parameters for model simulation (Supplementary Data 1).

### WQQM-PIDU model simulation representing different irrigation drainage systems

A water quantity and quality model for paddy irrigation and drainage units (WQQM-PIDU) was used to simulate water consumption by rice production and wastewater release. The WQQM-PIDU model was newly developed based on known knowledge of hydrology and nutrient cycling as well as two-year observations in China[26]. It simulates daily water quantity and water quality variations of a typical irrigation drainage unit (IDU) composed of fields, ditches (and a pond if existed). To represent freshwater irrigation supply from remote reservoirs, freshwater irrigation out of IDUs was simulated to be adequate in normal climate years, but was restricted in dry conditions beyond the supply capability of the irrigation district. First, model simulation was conducted on typical virtual IDUs with four different management styles (totally centralized, partly decentralized 1, partly decentralized 2, and quasi-decentralized) with surveyed IDU structure (Fig. 1c) and area percentages of ditches and ponds for different provinces (Fig. 2b). The results of the totally centralized system dominant currently and the quasi-decentralized one like the traditional system were compared for water sustainability (Fig. 3) and for system resilience (Fig. 4). Then, these IDU-scale results were randomly and proportionally sampled according to the percentages of these four sub-systems occurring in a region to generate provincial and regional results. For current systems, the expert survey data on the percentages of the four sub-systems were used for upscale sampling. For the proposed redesigned systems, 24 scenarios representing different implementation extent were used for upscale sampling. These scenarios represent the three proposed approaches of system redesign–recycling irrigation, pond reconnection, and pond construction–from easy to difficult implementation. The first set of scenarios represents the easily implemented approach that would increase the recycling irrigation percentages from the current status to 100%; the second set of scenarios represents the moderately easy-to-implement approaches that would further conduct pond reconnection to improve the percentage of existing ponds in IDUs to 100%; and the third set of scenarios additionally represent the difficult to implement the approach of pond construction to increase the area percentages of ditches and ponds to 8%, the maximum allowable percentage set by the general well-facilitated farmland construction standards of China (GB/T 30600-2022). The first two sets of scenarios were simulated by increasing the sampling percentages of irrigation drainage styles utilizing more small water bodies (Supplementary Table 3) while the ditch and pond parameters remained the same with the current status (Supplementary Table 4). The third set of scenarios were simulated by further increasing the area percentage of ponds in IDUs (Supplementary Table 5). All simulations were conducted for 30 years (1988–2017) of realistic daily climate data, and the results for analysis used the average value with probability ranges. We choose the WQQM-PIDU model for this study for its capability of

representing different irrigation drainage systems, but this model has not been widely used previously. The application of the model in more sites with local data would promote model development.

## System sustainability and resilience analysis

The sustainability of rice production was quantified by green, blue, and gray water footprint (WF) using the evapotranspiration and nutrient loads simulated by the WQQM-PIDU model. The model simulates daily crop evapotranspiration (ETc) using the FAO Penman-Monteith method multiplied by a crop coefficient[50]. The actual evapotranspiration ($ET_a$) is ETc multiplied by the water stress coefficient determined by the soil water as the method of FAO CROPWAT[29]. $ET_a$ from freshwater irrigation is calculated as blue water footprints and $ET_a$ from rainfall is calculated as green water footprints[28]. The WQQM-PIDU also simulates daily nutrient concentrations of fields, ditches, and ponds with input and output waters under fully mixed conditions and retention based on nutrient spiraling theory[51]. Then, nutrient flux from the IDU to surrounding waters was calculated as gray WF ($GWF_{runoff}$ via surface runoff *and* $GWF_{leaching}$ via subsurface leaching)[52] with the equations:

$$GWF_{runoff} = \frac{L_{runoff}*1000}{(c_{max,surface} - c_{nat}) \cdot Y} \tag{1}$$

$$GWF_{leaching} = \frac{L_{leaching}*1000}{(c_{max,groundwater} - c_{nat}) \cdot Y} \tag{2}$$

where $L_{runoff}$ and $L_{leaching}$ are the nutrient flux through runoff and leaching from paddy IDU respectively simulated by the WQQM-PIDU, in kg/ha; $c_{max,surface}$ and $c_{max,groundwater}$ are the maximum acceptable nutrient concentrations for receiving surface water and groundwater respectively, assigned to be 2 mg/L for nitrogen and 0.4 mg/L for phosphorus as grade V required by surface water quality standards of China (GB3838-2002), and 20 mg/L for nitrogen of groundwater as required by grade III of the groundwater quality standard of China (GB/T14848-1993)[53]; $c_{nat}$ is the natural concentration of nutrients in the receiving water-body, usually assumed to be 0[52]; $Y$ is the rice yield in ton/ha, the average annual value of statistical data during 2009-2018 for each province was used for calculation. The gray WF is calculated separately for nitrogen and phosphorus, and the final gray WF is the larger value calculated from them. Theoretically, four differently managed IDUs only changed the irrigation sources rather than the quantity of irrigation to paddy fields so that did not impact rice yield per unit area in normal climate years, while the irrigation self-sufficiency may reduce the risk of yield loss in extreme weather events. Hence, in the calculation of WF, the yield per unit area used for differently managed IDUs were conservatively assumed to be the same, while the field occupation effect by pond construction on total rice yield was considered in the calculation.

The resilience of the rice-production system to extreme weather is described by the local storage volume (LSV), the system irrigation self-sufficiency (*ISS*) and the potential yield loss due to water stress in dry conditions. The LSV is the volume of field drainage water that ditches and ponds within an IDU can store, in mm and is calculated with Eq. (3); the ISS is the percentage of irrigation water from local ditches and ponds to the total irrigation water needed for rice growth, in % and is calculated with Eq. (4):

$$LSV = \frac{V_{DP}*1000}{A_F} \tag{3}$$

$$ISS = \frac{I_{DP}*100}{I_{all}} \tag{4}$$

where $V_{DP}$ is the volume of ditches and ponds within an IDU, in m³; $A_F$ is the area of paddy fields, in m²; $I_{DP}$ is the irrigation water from ditches and ponds within an IDU, and $I_{all}$ is the total irrigation water needed for rice growth. The potential yield loss due to water stress caused by the reduction of freshwater irrigation was simulated only for dry conditions where the probability of irrigation (PI) was exceeded. The PI index represents the probability of remote freshwater supply can be guaranteed by reservoirs in various climate conditions. According to the well-facilitated farmland construction plan in China, the PI in rice production systems were set to be 79% for the Northeast rice region and Sichuan, Guizhou, Yunnan, and Guangxi provinces, 83% for other provinces in the Southeast coastal region and 87% for other provinces in the Changjiang River basin rice region. In dry climate conditions beyond the PI, freshwater irrigation was reduced and the potential yield loss (YL, the relative yield reduction, in %) was calculated using the approach of FAO CROPWAT[29] in which the relative yield reduction is related to the corresponding relative reduction in evapotranspiration as Eq. (5):

$$YL = \left(1 - \frac{Y_a}{Y_x}\right)*100 = K_y\left(1 - \frac{ET_a}{ET_c}\right)*100 \tag{5}$$

where $Y_a$ and $Y_x$ are actual and maximum yields, $ET_a$ and ETc are the actual and maximum crop evapotranspiration, and $K_y$ is a yield response factor representing the effect of a reduction in evapotranspiration on yield losses. $K_y$ was set to be 1.0, 1.09, 1.32, and 0.5 for the initial, development, mid-season, and late-season stages of the rice-growing season. It should be noted that this study only analyzed the benefit of yield stabilization under dry climate, while the benefit under wet climate was not quantified. This may underestimate the yield stabilization benefit since the additional small water bodies can also provide additional storage for field drainage, which can alleviate water-logging disasters.

## Robustness of model simulation

The robustness of the simulation results is guaranteed by both model calibration and uncertainty quantification. First, the model was calibrated by comparing the simulated nutrient export from paddy fields via runoff with published literature values by ref. 54. The general percent bias was 9.7% for nitrogen and 11.4% for phosphorus, and the coefficient of determination ($R^2$) for the provincial variations were 0.85 for nitrogen and 0.87 for phosphorus (Supplementary Fig. 3). Secondly, nutrient retention-related parameters of the WQQM-PIDU are the primary source of uncertainty. Therefore, 100 sets of these parameters (*vf_N*, *vf_P*, *ENCO*, and *EPCO*) generated from a log-normal probability distribution fitted by literature-reported values were used to quantify the uncertainty in terms of gray WFs as described above (Supplementary Fig. 2). These parameters represent a wide range of possible variability in reality (Supplementary Data 1). The provincial scale calibration is sufficient for the purpose of this study, which is to generate national and regional policy recommendations for rice irrigation drainage system transformation, rather than assist site-specific design. Finer-scale data and site-specific model calibration may be necessary to support the design of a specific rice irrigation drainage district or irrigation drainage unit in future studies.

## Cost and trade-off analysis of the proposed system redesign scenarios

The primary trade-off of the proposed system redesign approaches lies in the additional usage of ponds, the cost of land occupation and possible total rice yield loss is quantified for the pond construction approach. To address this trade-off, the survey and further analysis of the study only included small ponds of less than 0.33 ha (Fig. 2b), excluding ponds that are currently in use for aquaculture production or have unclear ownership. In addition, we quantified the field

occupation resulting from the pond construction approach, based on an assumption that all newly constructed ponds are converted from paddy fields, allowing for the possible total yield reduction due to the occupied fields to be quantified (as shown in Fig. 5).

## Reporting summary
Further information on research design is available in the Nature Portfolio Reporting Summary linked to this article.

## Data availability
Source data are provided with this paper. Statistical data on irrigation drainage system development in China are mainly from the National Bureau of Statistics (https://data.stats.gov.cn); climate data are from the China meteorological data service center (https://data.cma.cn). Other data supporting the main findings of this study and the important parameters used in the modeling analysis can be found in the Supplementary Information and the Supplementary Data files. Source data are provided with this paper.

## Code availability
The WQQM-PIDU model (v2.0) code and analysis scripts for water footprint and irrigation self-sufficiency calculation in this study are available on GitHub: https://github.com/Li-Sisi2020/WQQM-PIDU with a DOI[55]: https://zenodo.org/badge/latestdoi/538525282.

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

## Acknowledgements
This work was supported by the National Key Research and Development Program of China (2022YFD1700700, S.L., H.L., and L.Z.), the National Natural Science Foundation of China (U21A2025, L.Z. and 42207059, S.L.), and the Earmarked fund for the China Agriculture Research System (CARS-01-33, H.L.).

## Author contributions
S.L. and L. Zhang designed the study. Y.Z., F.Z., L. Zhai, X.F., M.L., and S.W. prepared the data. S.L., W.S., S.R., and M.M. analyzed the data and prepared the figures. S.L. wrote the paper and Z.W., L. Zhang, H.L. revised the paper. H.L., S.N., J.C., C.X., N.W., Y.D. contributed to discussing the results.

## Competing interests
The authors declare no competing interests.
