## [Peer Review File · Nature Communications]

Enhancing rice production sustainability and resilience via reactivating small water bodies for irrigation and drainageEditorial Note: This manuscript has been previously reviewed at another journal that is not operating a transparent peer review scheme. This document only contains reviewer comments and rebuttal letters for versions considered at *Nature Communications*. Mentions of prior referee reports have been redacted.

Reviewers' Comments:

Reviewer #1:

Remarks to the Author:

Food supply is essentially important for China, while facing great challenge with increasing demand and changing climate. A more sustainable and resilient irrigation drainage system can provide adaptive solutions to rice production. The authors propose a new pattern by reactivating small water bodies in irrigation and drainage to enhance rice production sustainability and resilience. It can decrease the water footprint of rice production by 30%, saving 9% of China's freshwater consumption and meanwhile increase the irrigation self-sufficiency from 3% to 31%, alleviating 2–3% yield loss in dry years. The findings provide helpful information and recommendations for agricultural and water resource management policies in China.

I read all the documents that the authors provided, especially the responses and the modified manuscript. Generally, I believe the authors have addressed most of the concerns in the previous comments.

//My comments on the authors' response to Reviewer #2's concerns:

The primary concern of Reviewer #2 was about the data and the modeling robustness. The authors have addressed the data sources at different spatial scales and how they are used for modeling. For the model calibration and validation, I agree with the authors' statement that the goal of the current study is to give national and regional policy recommendations; while I suggest that more site-specific validation (if more available data) or more sensitivity and robustness analysis will improve the modeling performance for national and regional analysis.

//

I (Reviewer #1) am still uncertain about little-used model WQQM-PIDU. My opinion is that, (a) firstly, we should encourage the researchers to develop their models since sometimes there are not fully available models to use; (b)while secondly, more efforts are needed to validate for the new models (can be referred to my comments in the last paragraph).

In conclusion, I believe that the current study proposed a very interesting topic and should be very important for the decision making on agricultural water resource management in China. The authors have done great job and improved the manuscript.

REVIEWERS' COMMENTS

Reviewer #1 (Remarks to the Author):

Food supply is essentially important for China, while facing great challenge with increasing demand and changing climate. A more sustainable and resilient irrigation drainage system can provide adaptive solutions to rice production. The authors propose a new pattern by reactivating small water bodies in irrigation and drainage to enhance rice production sustainability and resilience. It can decrease the water footprint of rice production by 30%, saving 9% of China's freshwater consumption and meanwhile increase the irrigation self-sufficiency from 3% to 31%, alleviating 2 - 3% yield loss in dry years. The findings provide helpful information and recommendations for agricultural and water resource management policies in China.

I read all the documents that the authors provided, especially the responses and the modified manuscript. Generally, I believe the authors have addressed most of the concerns in the previous comments.

//My comments on the authors' response to Reviewer #2's concerns:

The primary concern of Reviewer #2 was about the data and the modeling robustness. The authors have addressed the data sources at different spatial scales and how they are used for modeling. For the model calibration and validation, I agree with the authors' statement that the goal of the current study is to give national and regional policy recommendations; while I suggest that more site-specific validation (if more available data) or more sensitivity and robustness analysis will improve the modeling performance for national and regional analysis.

Response: Thank you for your acknowledgement on our previous revisions. For the model calibration and validation, we previously validate the model structure in a site with two-year observations (Li et al, 2020). For this study, we calibrate the model parameters in the provincial scales for the purpose of giving national and regional policy recommendations; as site-specific data is not applicable, we used uncertainty analysis to improve the modeling performance. First, we chose the most sensitive model parameters, i.e., nutrient retention related parameters, to do uncertainty quantification of the modelling results (lines 485-494 and Supplementary Fig. 2). 100 sets of these parameters (Supplementary Data 1) generated from a log-normal probability distribution fitted by literature reported values for ditches and ponds were used to quantify the uncertainty of nutrient loads and gray water footprint simulation. In addition, we used 90 sets of climate data representing three typical sites in a province and 30 years of realistic climate data to run the model. Hence, all related results have a probability range representing climatic and parameter variability of the realistic condition in China (Figure 3, 5), which ensures the robustness of the modeling results.

//

I (Reviewer #1) am still uncertain about little-used model WQQM-PIDU. My opinion is that, (a)

firstly, we should encourage the researchers to develop their models since sometimes there are not fully available models to use; (b)while secondly, more efforts are needed to validate for the new models (can be referred to my comments in the last paragraph).

Response: Thank you for your encouragement for newly developed models like this, we did robustness analysis for this study and added some discussion to state the model used in this study has not been widely used previously. Meanwhile, we offer the model source code and invite more researchers to use and test it for the development of models. (Lines 417-420)

In conclusion, I believe that the current study proposed a very interesting topic and should be very important for the decision making on agricultural water resource management in China. The authors have done great job and improved the manuscript.

Response: Thank you for your comment. In the revised manuscript, we toned down our claims of generalization of our findings for the globe in the discussion section and focus on summarizing its importance for the decision making in China. (Lines 298, 316-321, 334-336)